# Association of Sleep Duration and Self-Reported Insomnia Symptoms with Metabolic Syndrome Components among Middle-Aged and Older Adults

**DOI:** 10.3390/ijerph191811637

**Published:** 2022-09-15

**Authors:** Yuting Zhang, Yingcai Xie, Lingling Huang, Yan Zhang, Xilin Li, Qiyu Fang, Qun Wang

**Affiliations:** 1Health Science Centre, Shenzhen University, Shenzhen 518060, China; 2Department of Endocrinology, Affiliated Hospital of Guangdong Medical University, Zhanjiang 524000, China

**Keywords:** daily sleep duration, insomnia symptoms, metabolic syndrome, cross-sectional study

## Abstract

The study aimed to explore the association between sleep duration, insomnia symptoms and the components of metabolic syndrome (MetS) among middle-aged and older adults. A cross-sectional study was conducted in five community health centers and physical check-up centers of two comprehensive hospitals in Guangdong. We recruited 1252 participants (658 female), aged 40–96 years and with a body mass index (BMI) of 16.26–35.56 kg/m^2^. MetS was assessed based on the guidelines of the International Diabetes Federation. Self-reported sleep duration was evaluated by a simplified questionnaire. Compared with the participants who slept 6–8 h/day, those who slept shorter (<6 h/day) or longer (>8 h/day) periods of time with or without insomnia symptoms had significantly increased odds ratios (ORs) of high blood pressure (except for the SBP in model 2) and high triglycerides (TGs) in all models (*p* < 0.05), whereas the participants who slept longer (>8 h/day) or shorter (<6 h/day) periods of time with insomnia symptoms had significantly increased ORs of low HDL-C in all models (*p* < 0.05), but non-significant in those without insomnia symptoms. BMI is significant for insomnia symptoms but not for sleep duration. Our study indicated that the association of sleep duration with MetS components was partially associated with insomnia symptoms. These findings have significant implications to explore the appropriate sleep duration for adults.

## 1. Introduction

The metabolic syndrome (MetS) is a combination of metabolic abnormalities including hypertension, dyslipidemia, abdominal obesity and disturbed glucose and insulin metabolism [1], all of these being well associated with high-risk factors for both type 2 diabetes mellitus (T2DM) and cardiovascular diseases (CVDs) [2,3,4]. The prevalence of MetS has considerably increased over the past few years and is now at epidemic proportions in many countries [5,6,7,8]. In China, the prevalence of MetS among the age-standardized group was reported as 17.8% in females and 9.8% in males [9,10]. A previous study indicated that MetS is linked to the developing of T2DM and its complications [11]. In addition, a recent survey indicated that the prevalence of MetS in the T2DM groups was 73.5% in males and 75.4% in females [12].Therefore, growing evidence suggests that it is essential to identify the modifiable risk factors to delay the progression of MetS.

Sleep duration, one of the traditional important modifiable risk factors, has been proved to be closely related to human MetS health [13]. Numerous studies have indicated that short sleep duration had negative effects on MetS, including hypertension [14,15], diabetes [16], obesity [17] and escalated mortality [18]. However, the findings are not consistent. In the current existing literature, there is also evidence of possible associations between short or long sleep duration and a risk of MetS [18,19]. Although a considerable number of studies have indicated that both quantitative and qualitative sleep disturbances are associated with a risk of MetS and its components [20,21], few studies focus on the individuals with metabolic syndrome. Furthermore, insomnia, which is increasingly prevalent in modern society, is associated with sleep disorders. The prevalence of insomnia ranges between 8% and 40% all over the world [2], whereas the association between MetS and insomnia symptoms remains unclear. Numerous studies have revealed that self-reported symptoms of insomnia were associated with MetS [2,22], while others did not [23,24]. Moreover, to our knowledge, epidemiological studies exploring the association of sleep duration, insomnia symptoms and MetS in conjunction with the components are scarce.

To fill the above-mentioned knowledge gap, we performed a cross-sectional study to explore the association of sleep duration and insomnia symptoms with MetS components among middle-aged and older adults in Guangdong.

## 2. Materials and Methods

### 2.1. Data Source and Volunteers

From March 2021 to May 2022, we recruited a convenience sample of 1252 adults aged 40 years or older from 5 community health centers and physical check-up centers of 2 comprehensive hospitals in Guangdong (China). All participants received a routine body check-up including anthropometry (such as weight, height, waistline circumference), blood pressure, venous blood sampling of triglycerides (TGs), high-density lipoprotein-cholesterol (HDL-C), fasting blood glucose (FBG) and 2-h postprandial blood Glucose (2hPG). Written informed consent was obtained from all individuals before participating.

The study was approved by the Ethics Committee of Health Science Centre, Shenzhen University. MetS was defined based on the National Cholesterol Education Program Adult Treatment Panel III guidelines (NCEP/ATPIII), updated by the American Heart Association and the National Heart, Lung and Blood Institute in 2005 [25].

### 2.2. Definitions of MetS

MetS was defined as participants who met at least three of the following criteria: (1) systolic (SBP) and/or diastolic blood pressure (DBP) ≥130 mmHg/85 mmHg, or taking antihypertensive medication; (2) HDL-C < 1.29 mmol/L in women, <1.03 mmol/L in men or particular medication for dyslipidemia; (3) FBG ≥ 5.60 mmol/L or previously diagnosed T2DM; (4) TG ≥ 1.70 mmol/L or particular medication for dyslipidemia; (5) central obesity: WC ≥ 102 cm in men, ≥80 cm in women.

### 2.3. Data Collection

When attending body check-ups in the physical examination centers, a single item questionnaire of “How many hours do you sleep a day?” was used to measure individuals’ sleep duration, and the response was classified as “<4 h, 4–6 h, 6–8 h, 8–10 h or >10 h”. In our current survey, the response of “>10 h/day” was scarce; thus, we combined the data with those answered as “8–10 h/day”. Similarly, we adopted a self-administered questionnaire to acquire information regarding insomnia symptoms [2]. Participants were asked “How was your sleep quality last month?” with the following response options: 1 = difficulty initiating asleep, 2 = difficulty maintaining sleep, 3 = feeling of non-restorative sleep, 4 = taking sleeping pills, 5 = suffering from sleep apnea and 6 = sleeping well. According to the guidelines of the American Psychiatric Association (APA), participants who chose one of the former five above-mentioned options were regarded as “having insomnia symptoms”, while those who chose the last option were regarded as “not having insomnia symptoms” [2].

Anthropometric and biochemical assessments were performed by local professional medical staff. Participants followed relevant provisions of biochemical measurement, including fasting for 12–14 h and no strenuous exercise prior to blood being drawn. Weight and height were measured by an automatic anthropometer (Nakamura KN-5000A, Tokyo, Japan). BMI was calculated as weight in kilograms divided by height in meters, squared (kg/m^2^). WC was assessed at enrollment and measured at the narrowest point between the bottom of the ribs and the iliac crest. Blood pressure was measured by a sphygmomanometer (Yu yue, YJ100002, Zhenjiang, China) on the right arm after the patient had been supine for 20 min or longer. Levels of fasting plasma glucose (FBG), 2-h post-load glucose (2hPG), total cholesterol (TC), serum triglycerides (TGs) and high-density lipoprotein cholesterol (HDL-C) were enzymatically assessed using a semi-automated analyzer (Sysmex XN-3000, Tokyo, Japan).

### 2.4. Covariates

Demographic information of participants was collected through face-to-face interviews. Personal information consisted of gender (male and female), age, marital status (not married, married and divorced), level of education (high school and below, college and universities, master and PhD), current smoking and drinking and status, breakfast habits, physical activity, sitting time and insomnia symptoms. Insomnia symptoms were dichotomized as yes and no. Breakfast habits were categorized as none, 1–3 times/week, 4–5 times/week and every day. Physical activity was categorized into four groups: <0.5 h, 0.5–1 h, 1–2 h, >2 h. Sitting time was categorized as <6 h, 6–8 h, 8–10 h, >10 h.

### 2.5. Statistical Analysis

All variables were expressed as mean (standard deviation) for continuous data and percentages for categorical variables according to the Shapiro–Wilk test of normality. For continuous variables, a Wilcoxon–Mann–Whitney test was performed. For categorical variables, a Chi-squared test with Yates’ continuity correction was conducted. A stepwise multivariate-adjusted logistic regression analysis was conducted to investigate the association, and the 95% confidence intervals (CIs) and odds ratios (ORs) and were determined. A stratified analysis by insomnia symptoms was conducted to estimate the effect of insomnia symptoms on sleep duration and the MetS components. We controlled covariates including gender, age, marital status, level of education, current drinking and smoking status, physical activity and sitting time for all the regression models. A two-sided *p* value < 0.05 was inferred as statistical significance, and *SPSS* software (version 24.0, IBM Corp, Chicago, IL, USA) was used to conduct analyses.

## 3. Results

The general characteristics of the participants are presented in Table 1. Among 1252 participants (52.56% female and 47.44% male) with MetS, 16.93% of participants slept <6 h/day, 54.63% slept 6–8 h/day and 28.43% slept >8 h/day. Over half of the participants (n = 865, 69.09%) had insomnia symptoms, and 39.14% (n = 490), 61.34% (n = 768), 37.22% (n = 466) and 27.80% (n = 348) of the participants had elevated BP, elevated blood glucose, high TG and high FBG, respectively. Participants who slept <6 h/day or >8 h/day had a significantly higher BMI (26.42 ± 3.29 and 23.11 ± 3.38 kg/m^2^ versus 22.56 ± 3.23 kg/m^2^), SBP (139.50 ± 17.34 and 134.30 ± 18.80 mmHg versus 132.61 ± 17.78 mmHg), DBP (89.77 ± 12.73 and 87.97 ± 14.24 mmHg versus 85.59 ± 14.12 mmHg), TG (1.83 ± 0.48 and 1.63 ± 0.50 mmol/L versus 1.45 ± 0.36 mmol/L), and FBG (5.77 ± 0.44 and 5.77 ± 0.39 mmol/L versus 5.69 ± 0.62 mmol/L) than those who slept 6–8 h/day. Meanwhile, compared with those who slept 6–8 h/day, participants who slept <6 h/day or >8 h/day had significantly lower HDL-C (1.12 ± 0.24 and 1.25 ± 0.23 mmol/L versus 1.30 ± 0.27 mmol/L). Dietary intake and physical activity habits differences among the participants with different daily sleep durations are presented in Table 1.

Table 2 summarizes the ORs of the components of MetS across sleep duration. It shows a U-shaped relationship between the components of MetS and sleep duration. Compared with those who slept 6–8 h/day or >8 h/day, the participants who slept <6 h/day had a significantly increased ORs of abnormal BMI (except in Model 3), high BP (except for high SBP and DBP in Model 3), high TG and low HDL-C in all models (*p* < 0.05).

Table 3 showed the ORs of the components of MetS across volunteers with insomnia symptoms. Compared with those without insomnia symptoms, the participants with insomnia symptoms had significantly increased ORs of high abnormal BMI, high BP and high TG in all models (*p* < 0.05).

Lastly, a conducted stratified analysis was performed to explore whether insomnia symptoms affected the association of the components of MetS with sleep duration, as presented in Table 4. The ORs for the components of MetS also explained a U-shaped association with sleep duration in participants with or without insomnia symptoms. Compared with the participants who slept 6–8 h/day, those who slept <6 h/day with or without insomnia symptoms had significantly increased ORs of high blood pressure (except for SBP in model 2) and high TG in all models (*p* < 0.05), whereas the participants who slept >8 h/day or <6 h/day with insomnia symptoms had significantly increased ORs of low HDL-C in all models (*p* < 0.05), which were non-significant for those without insomnia symptoms. Thus, the association of sleep duration with MetS components was not affected by the insomnia symptoms.

## 4. Discussion

In this cross-sectional epidemiological survey, we examined the associations between sleep duration, self-reported insomnia symptoms and the components of MetS. The results revealed that short or long sleep had significantly increased ORs of high BP and high TG. In addition, among those with self-reported insomnia symptoms, short or long sleep were significantly associated with low HDL-C for both men and women. Furthermore, these associations were robust in the stratified analyses. The current findings complement previous existing studies on sleep duration and MetS by the simultaneous use of information on insomnia symptoms.

In the present study, a U-shaped association was revealed between individuals who slept less than 6 h daily or over 8 h daily, and MetS components compared with those who slept between 6 and 8 h a day. The findings supported a U-shaped curvilinear relationship between daily sleep duration and the risk of MetS [2,26]. Furthermore, short or long sleep duration and insomnia symptoms appeared to be significantly associated with the components of MetS, including abnormal BMI, blood pressure, triglyceride and HDL-C. These findings were also applicable to the participants with insomnia symptoms. The previous studies revealed that short sleep duration was associated with MetS [27]. However, there are also opposite epidemiological findings; a meta-analysis study indicated that individuals who slept less than 6–7 h daily were at a higher risk of MetS than those who slept between 7 and 8 h daily [26].

These inconsistent findings might be explained by discrepancies in sleep duration assessments and definitions. Our study assessed sleep duration based on a 24 h sleep recall data, whereas other studies measured sleep duration by systematic validated sleep questionnaires [28,29]. In addition, some studies defined sleep duration based only on nighttime sleep [2], including the present study, whereas some others combined both nighttime sleep and daytime napping as sleep duration [30].

As daytime napping has become more prevalent and the timing of sleep more variable, it is difficult to define sleep duration [31]. Meanwhile, different conclusions have been made on the effect of daytime napping on MetS and other diseases. Several studies demonstrated that longer daytime napping was associated with a higher risk of MetS [12,32], T2DM [33,34], abdominal obesity [35], renal hyperfiltration [36], diabetes mellitus [37,38] and major cardiovascular events [39], while others indicated that long daytime napping tended to be independent of diabetes, hypertension and obesity [36,40].

We found that insomnia symptoms were significantly associated with MetS components including high DBP, high TG and low HDL-C. Consistent with our findings, other studies also revealed that insomnia symptoms, including difficulty initiating asleep, difficulty maintaining sleep, feeling of non-restorative sleep, taking sleeping pills, and suffering from sleep apnea, were significantly associated with MetS [22,41]. A meta-analysis of epidemiological evidence indicated that sleep fragmentation, fatigue, anxiety and depressive symptoms might be comorbidity and confounding factors which could accelerate the development of MetS [42]. Another meta-analysis of epidemiological studies suggested that compared with individuals without insomnia, those insomnia patients suffered more from obesity, hypertension, hyperlipidemia and hyperglycemia in MetS [24]. Additionally, other epidemiological studies also indicated that MetS was closely associated with insomnia symptoms and other sleep disorders, no matter the race and gender [43,44]. This current study reveals that after adjusting for confounding variables, the relationship between daily sleep duration and MetS components was not modified by insomnia symptoms, indicating that insomnia symptoms could consistently be associated with MetS. The current report complements the available literature on sleep duration and MetS by the simultaneous information of insomnia symptoms and sleep duration.

The biological plausibility pathways of the relationship of insomnia symptoms with MetS remains unclear. Vgontzas [45] suggested that objective sleepiness is primarily related to metabolic factors, whereas fatigue appears to be related to psychological distress. The interaction of the hypothalamic–pituitary–adrenal (HPA) axis responding to insomnia and proinflammatory cytokines could lead to the level of sleep efficiency and objective sleepiness [45]. In addition, insulin resistance sensitivity might be decreased by sleep restriction or sleep fragmentation [46], which play a crucial role in the prevention of MetS-related brain changes [47]. Our stratified analysis reveals that differences in the relationship of sleep duration and MetS components between the participants with or without MetS, indicating that insomnia symptoms significantly influence the relationship between sleep duration and MetS components.

To the best of our knowledge, there is limited epidemiological evidence on the relationship of sleep duration, self-reported insomnia symptoms and MetS components among adults aged 40 years and over. Moreover, our study includes a comparable sample size and available blood measurements from the population of interest, which not only allows for multiple variables adjustment but also could provide more reliable estimates. However, there were also certain limitations which should be noted. First, the cross-sectional design could not establish any causal relationships among the variables, and possible reverse causation could not be fully excluded. Further studies should be conducted to clarify the mechanisms underlying the association among daily sleep duration, insomnia symptoms, MetS and its components. Second, instead of an objective actigraphic or polysomnographic instrument, we adapted a single self-reported questionnaire to measure sleep duration, which may lead to sleep recall bias. Nonetheless, the previous studies revealed that both self-reported and objective measurements of sleep quality were significantly associated with an increased risk of MetS [48,49]. Third, although we controlled for some confounding variables including age, gender, current drinking and smoking status, we lacked relevant information about sleep apnea, which might play a role in the relationship of sleep quality with MetS. Therefore, the possibility of other residual confounding variables cannot be excluded.

## 5. Conclusions

In conclusion, we found that short or long sleep duration and insomnia symptoms were significantly associated with a higher risk of MetS components. These findings should be further verified in prospective research using both sleep quality and sleep duration objective measures.

## Figures and Tables

**Table 1 ijerph-19-11637-t001:** Characteristics of the participants across sleep duration (N = 1252).

Variables	Sleep Duration (Hours/Day)			
	<6 (n = 212)	6–8 (n = 684)	>8 (n = 356)	*p*-Value
Gender (%)				0.505
Male	44.81	47.08	49.72	
Female	55.19	52.92	50.28	
Age (years)	61.58 ± 15.12	59.73 ± 16.59	62.03 ± 13.03	0.048
Marital status (%)				0.033
Not married	28.77	35.09	25.84	
Married	63.21	58.63	67.13	
Divorced	8.02	6.29	7.02	
Level of education (%)				0.408
High school and below	15.57	16.52	20.51	
College and universities	56.60	56.87	54.21	
Master	24.53	24.27	24.16	
PhD	3.30	2.34	1.12	
Current drinking status				<0.001
Yes	61.79%	39.91%	53.93%	
No	38.21%	60.09%	46.07%	
Current smoking status				<0.001
Yes	54.25%	41.08%	55.90%	
No	45.75%	58.92%	44.10%	
Breakfast habits (per week)				<0.001
No	4.72%	3.51%	4.21%	
1–3 times	41.04%	27.92%	32.87%	
4–5 times	31.13%	30.56%	39.89%	
Everyday	23.11%	38.01%	23.03%	
Physical activity (per day)				0.000
<0.5 h	50.00%	40.06%	24.72%	
0.5–1 h	25.94%	38.01%	24.72%	
1–2 h	13.68%	15.06%	29.21%	
>2 h	10.38%	6.87%	21.35%	
Sitting time (per day)				<0.001
<6 h	26.42%	24.85%	24.72%	
6–8 h	35.85%	41.81%	28.65%	
8–10 h	17.45%	21.05%	34.27%	
>10 h	20.28%	12.28%	12.36%	
Insomnia symptoms				0.000
Yes	79.25%	57.31%	85.67%	
No	20.75%	42.69%	14.33%	
Body mass index (kg/m2)	26.42 ± 3.29	22.56 ± 3.23	23.11 ± 3.38	<0.001
Systolic BP (mmHg)	139.50 ± 17.34	132.61 ± 17.78	134.30 ± 18.80	<0.001
Diastolic BP (mmHg)	89.77 ± 12.73	85.59 ± 14.12	87.97 ± 14.24	<0.001
Triglycerides (mmol/L)	1.83 ± 0.48	1.45 ± 0.36	1.63 ± 0.50	<0.001
HDL-C (mmol/L)	1.12 ± 0.24	1.30 ± 0.27	1.25 ± 0.23	<0.001
FBG (mmol/L)	5.77 ± 0.44	5.69 ± 0.62	5.77 ± 0.39	0.020
2hPG (mmol/L)	9.05 ± 0.70	9.09 ± 0.64	9.07 ± 0.69	0.789

**Table 2 ijerph-19-11637-t002:** Odds ratios (95% CI) of the components of metabolic syndrome and sleep duration (N = 1252).

Variables	Sleep Duration (Hours/Day)			
	<6 (n = 212)	6–8 (n = 684)	>8 (n = 356)	*p*-Trend
Abnormal BMI				
Model 1 ^1^	5.355 (3.725–7.696)	1	1.138 (0.879–1.475)	0.023
Model 2 ^2^	4.969 (3.439–7.179)	1	1.176 (0.893–1.547)	0.015
Model 3 ^3^	4.743 (3.274–6.873)	1	1.117 (0.845–1.476)	0.061
High systolic BP				
Model 1 ^1^	2.487 (1.760–3.513)	1	1.476 (1.134–1.922)	0.000
Model 2 ^2^	2.255 (1.572–3.237)	1	1.170 (0.880–1.557)	0.083
Model 3 ^3^	2.108 (1.463–3.038)	1	1.091 (0.815–1.459)	0.261
High diastolic BP				
Model 1 ^1^	3.409 (2.386–4.872)	1	1.552 (1.196–2.015)	0.000
Model 2 ^2^	3.166 (2.191–4.577)	1	1.315 (0.994–1.739)	0.005
Model 3 ^3^	2.861 (1.969–4.157)	1	1.171 (0.879–1.559)	0.068
High triglycerides				
Model 1 ^1^	12.060 (8.339–17.441)	1	2.600 (1.971–3.430)	0.000
Model 2 ^2^	11.853 (8.135–17.270)	1	2.334 (1.744–3.123)	0.000
Model 3 ^3^	11.779 (8.051–17.234)	1	2.319 (1.723–3.121)	0.000
Low HDL-C (men)				
Model 1 ^1^	9.649 (5.743–16.213)	1	1.797 (1.120–2.883)	0.001
Model 2 ^2^	10.263 (5.950–17.702)	1	2.087 (1.260–3.457)	0.000
Model 3 ^3^	9.525 (5.488–16.530)	1	1.942 (1.165–3.237)	0.001
Low HDL-C (women)				
Model 1 ^1^	2.524 (1.639–3.887)	1	1.789 (1.246–2.568)	0.000
Model 2 ^2^	2.443 (1.566–3.811)	1	1.639 (1.114–2.411)	0.003
Model 3 ^3^	2.427 (1.550–3.801)	1	1.625 (1.096–2.409)	0.004

The ORs across categories of sleep duration were compared with the reference group (6–8 h/day). The variables were defined as abnormal BMI (mean): <18.5 and ≥24.0, high SBP: 2130 mmHg, high DBP: 285 mmHg, low HDL-C: <1.03 mmol/L for men and <1.29 mmol/L for women, high TG: 21.70 mmol/L. ^1^ Unadjusted. ^2^ Adjusted for gender, age, marital status, level of education, current drinking and smoking status, physical activity and sitting time. ^3^ Adjusted for gender, age, marital status, level of education, current drinking and smoking status, physical activity, sitting time and insomnia symptoms.

**Table 3 ijerph-19-11637-t003:** Odds ratios (95% CI) of the components of MetS and insomnia symptoms. ORs: odds ratios (N = 1252).

Variables	Volunteers with Insomnia Symptoms OR (95% CI)	*p*-Trend
Abnormal BMI		
Model 1 ^1^	1.451 (1.139–1.849)	0.003
Model 2 ^2^	1.511 (1.156–1.975)	0.003
Model 3 ^3^	1.580 (1.203–2.074)	0.001
High systolic BP		
Model 1 ^1^	2.077 (1.627–2.651)	0.000
Model 2 ^2^	1.565 (1.190–2.058)	0.001
Model 3 ^3^	1.593 (1.209–2.097)	0.001
High diastolic BP		
Model 1 ^1^	2.430 (1.902–3.104)	0.000
Model 2 ^2^	2.072 (1.579–2.719)	0.000
Model 3 ^3^	2.130 (1.620–2.802)	0.000
High triglycerides		
Model 1 ^1^	1.701 (1.314–2.202)	0.000
Model 2 ^2^	1.448 (1.092–1.920)	0.010
Model 3 ^3^	1.507 (1.132–2.007)	0.005
Low HDL-C (men)		
Model 1 ^1^	1.662 (1.080–2.559)	0.021
Model 2 ^2^	2.056 (1.285–3.290)	0.003
Model 3 ^3^	2.054 (1.276–3.307)	0.003
Low HDL-C (women)		
Model 1 ^1^	1.470 (1.051–2.055)	0.024
Model 2 ^2^	1.204 (0.828–1.750)	0.331
Model 3 ^3^	1.222 (0.840–1.779)	0.295

^1^ Unadjusted. ^2^ Adjusted for sex, age, marital status, level of education, current drinking and smoking status, physical activity and sitting time. ^3^ Adjusted for sex, age, marital status, level of education, current drinking and smoking status, physical activity, sitting time and sleep duration.

**Table 4 ijerph-19-11637-t004:** Odds ratios (95% CI) of the components of metabolic syndrome across participants with or without insomnia symptoms (N = 1252).

Variables	Participants without Insomnia Symptoms				Participants with Insomnia Symptoms			
	Sleep Duration (Hours/Day)				Sleep Duration (Hours/Day)			
	<6	6–8	>8	*p*-Trend	<6	6–8	>8	*p*-Trend
Abnormal BMI								
Model 1 ^1^	6.250 (2.895–13.493)	1	0.670 (0.351–1.279)	0.571	4.889 (3.214–7.436)	1	1.162 (0.860–1.570)	0.160
Model 2 ^2^	5.961 (2.675–13.283)	1	0.614 (0.311–1.212)	0.882	4.667 (3.054–7.131)	1	1.241 (0.906–1.700)	0.054
High SBP								
Model 1 ^1^	2.181 (1.131–4.203)	1	1.780(0.974–3.255)	0.015	2.148 (1.414–3.262)	1	1.122 (0.822–1.533)	0.349
Model 2 ^2^	2.516 (1.156–5.476)	1	1.605 (0.813–3.170)	0.057	2.113 (1.376–3.246)	1	1.020 (0.734–1.418)	0.660
High DBP								
Model 1 ^1^	2.276 (1.188–4.361)	1	1.003 (0.548–1.836)	0.415	3.344 (2.147–5.206)	1	1.345 (0.987–1.832)	0.027
Model 2 ^2^	2.554 (1.184–5.506)	1	0.810 (0.413–1.588)	0.906	3.371 (2.147–5.291)	1	1.303 (0.941–1.805)	0.038
High TG								
Model 1 ^1^	8.115 (4.053–16.248)	1	2.651 (1.419–4.952)	0.000	13.225 (8.484–20.615)	1	2.493 (1.798–3.457)	0.000
Model 2 ^2^	9.760 (4.562–20.878)	1	2.664 (1.359–5.221)	0.000	13.408 (8.545–21.038)	1	2.373 (1.693–3.325)	0.000
Low HDL-C (men)								
Model 1 ^1^	5.950 (2.197–16.114)	1	0.952 (0.299–3.027)	0.323	11.236 (5.968–21.153)	1	2.031 (1.155–3.573)	0.012
Model 2 ^2^	6.436 (2.182–18.988)	1	1.106 (0.325–3.768)	0.267	11.889 (6.102–23.164)	1	2.333 (1.295–4.203)	0.002
Low HDL-C (women)								
Model 1 ^1^	3.869 (1.514–9.890)	1	1.103 (0.444–2.741)	0.204	2.149 (1.302–3.545)	1	1.788 (1.176–2.717)	0.004
Model 2 ^2^	4.216 (1.445–12.301)	1	1.014 (0.380–2.707)	0.361	2.131 (1.281–3.545)	1	1.744 (1.123–2.709)	0.007

^1^ Unadjusted. ^2^ Adjusted for sex, age, marital status, level of education, current drinking and smoking status, physical activity and sitting time.

## Data Availability

The data presented in this study are available upon request from the corresponding author. The data are not publicly available due to privacy restrictions.

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
