# Peer review of "Association of Sleep Duration and Self-Reported Insomnia Symptoms with Metabolic Syndrome Components among Middle-Aged and Older Adults"

_ijerph, 2022, doi:10.3390/ijerph191811637_

Round 1

Reviewer 1 Report

Major

1. Line 85. Is there sufficient evidence to regard "suffering from sleep apnea" as "with insomnia"?  Is it possible to increase the proportion of subjects with insomnia?

    On the other hand, some participants did not know whether they had OSAHS. Self-reported "suffering from sleep apnea"  may lead to underestimate the number of subjects "with insomnia" without PSG examination.

Further more, as the diagnosis of insomnia is solely based on subjective complaints, if the 6 problems were accuray for the diagnosis of insomnia? Is there any basis? Please list in the citation. 

2. Line 97. If the difference of blood pressure between the two upper limbs of the subject is greater than 20 mmHg, the higher side is taken generally.  In this study, the blood pressure was measured only on right arm, which maybe lead underestimation of  the hypertension in some subjects.

3. In Table 1, the statistical results of blood pressure comparison among the three subgroups with different sleep time (P < 0.05) could only indicate that the blood pressure of the three subgroups was not completely similar, but cannot indicate that the blood pressure of the subgroups with sleep time <6h and ≥8h was higher than that of the subgroups with 6-8h sleep time. Multiple comparisons between subgroups were required to draw the conclusion.

4. Line 230, according to " the relationship between daily sleep duration and MetS components was not modified by insomnia symptoms",  It can only be explained as sleep time is related to MetS, and it seems that we can not conclude that insomnia affects MetS.

5. In order to show the U-shaped correlation between the parameters, it is suggest to convert the data table into graphic format.

Minor

1. Please pay attention to spelling mistakes, for example, the left bracket is missing on Line 133, etc. 

2. It is necessary to revise according to the native English, and pay attention to grammar.

Author Response

  On behalf of my co-authors, we are very grateful to you for giving us the opportunity to revise our manuscript. We appreciate you very much for your positive and constructive comments and suggestions on our manuscript “Association of Daily Sleep Duration and Insomnia Symptoms with Metabolic Syndrome and its Components among Adults in Guangdong” (ID: ijerph-1867424).

 We have studied reviewers’ comments carefully and tried out best to revise our manuscript according to the comments. The following are the responses and revisions I have made in response to the reviewers’ questions and suggestions on an item-by-item basis. Thanks again to the hard work of the editor and reviewers.

Reviewer 2 Report

Congratulations to the authors for taking up the topic and for efficiently conducted research.

Below are my comments.

1.       Introduction

 It would be worthwhile for the authors to elaborate a little more on the introductory part of the article. Please expand the theoretical part a bit. for example about articles:

https://doi.org/10.3390/ijerph191610368

https://doi.org/10.3390/ijerph191610250

https://doi.org/10.3390/ijerph191610203

https://doi.org/10.3390/antiox11081573

https://doi.org/10.3390/jcm11164751

https://doi.org/10.3390/biomedicines10081957

Materials and Methods and 3. Results

This part is correct and exhaustive description, no need to make any changes

4. Discussion

Basically, the authors refer to relevant other studies correctly interpreting the obtained results. The conclusions are correctly derived and confronted with the research carried out by other authors.

Author Response

(The authors gave the same response as above.)

Reviewer 3 Report

Major issues

The age rage is too wide (18-96 yo), considering that the MetS appears at middle age.

Insulin resistance parameters should be tested, since are related to MetS.

FBG is significant in Table 1, but the analysis is not made subsequently.

Since the results are different for insomnia and sleep duration, conclusions should be drawn separately. (For instance BMI is significant for insomnia but not for sleep duration)

HDL-C is not significant for women (table 3), and not depends on sleep duration (table 4).

The criteria to include the patients as MetS affected are 3 of 5 symptoms,. However, only one and a half are affected by sleep duration and insomnia. For these reasons, could not be concluded that the sleep disorders affect MetS, but only these ywo symptoms.

Minor issues

Table headers must be on the same page as the table.

The manuscript shows 23% of coincidence with ref 2 (Syauqui et al 2019).

Author Response

(The authors gave the same response as above.)

Reviewer 4 Report

The paper entitled “Association of Daily Sleep Duration and Insomnia Symptoms with Metabolic Syndrome and its Components among Adults in Shenzhen” is very well structured and presented. The main topics of the paper are well introduced and the methodology and results are well organized. This work showed the importance of the sleep duration and its association with metabolic syndrome components. The information is clear, and the authors also correctly discussed the limitations of the study. The statistical analysis is adequate.

Author Response

(The authors gave the same response as above.)

Round 2

Reviewer 3 Report

The author has not fixed all the issues from the previous review round.

The similarity is still 25% with the ref. 2.

The title and the conclusions have not been duly modified. MetS is not related with  both sleep duration and insomnia, but just some symptoms , and not always with sleep duration and insomnia together.

Author Response

  We are very grateful to you for giving us the opportunity to revise our manuscript. We appreciate you very much for your positive and constructive comments and suggestions on our manuscript (ID: ijerph-1867424).

 We have studied reviewers’ comments carefully and tried out best to revise our manuscript according to the comments. The following are the revisions I have made in response to the reviewers’ questions and suggestions on an item-by-item basis. Thanks again to the hard work of the editor and reviewers.

  1. We changed the title according to the reviewer’s suggestion.
  2. All revisions made to the manuscript were marked as “Red”,
  3. We checked all references are relevant to the contents of the manuscript.
  4. Actually, we do use insomnia “symptoms” in our manuscript, we dentally expressed the single item to measure sleep insomnia.
  5. It is true that “MetS” was not always with sleep duration and insomnia together, we discussed the finding case by case in our discussion part.

Special thanks to all your valuable advice.